# LEARNING TEST TIME AUGMENTATION WITH CASCADE LOSS PREDICTION

## ABSTRACT

Data augmentation has been a successful common practice for improving the performance of deep neural network during training stage. In recent years, studies on test time augmentation (TTA) have also been promising due to its effectiveness on improving the robustness against out-of-distribution data at inference. Instead of simply adopting pre-defined handcrafted geometric operations such as cropping and flipping, recent TTA methods learn predictive transformations which are supposed to provide the best performance gain on each test sample. However, the desired iteration number of transformation is proportional to the inference time of the predictor, and the gain by ensembling multiple augmented inputs still requires additional forward pass of the target model. In this paper, we propose a cascade method for test time augmentation prediction. It only requires a single forward pass of the transformation predictor, while can output multiple desirable transformations iteratively. These transformations will then be adopted sequentially on the test sample at once before the target model inference. The experimental results show that our method provides a better trade-off between computational cost and overall performance at test time, and shows significant improvement compared to existing methods.

## 1 INTRODUCTION

Robustness in artificial intelligence system has been recognized as an important topic in recent years, especially for the application scenario closely related to human life or health, such as biometrics, autonomous driving, medical diagnosis, virtual and augmented reality and so on. Though heavily rely on training data, AI models in real-world will inevitably encounter unforeseen circumstances, which requires not only high performance from the aspect of accuracy, but also high robustness from the aspect of generalization.

Data augmentation has been a successful strategy for improving the robustness in many deep learning model training applications. During training stage, various transformations are adopted on the input samples thus to expend the diversity of the training space without truly collecting novel data. In the community of computer vision, some basic augmentation operations are commonly used such as rotation, zoom-in and -out, cropping, flipping, translation, blur, contrast, etc. Some advanced techniques also explore sub-instance level operations such as mixing samples together (Zhang et al., 2018; DeVries & Taylor, 2017; Hendrycks et al., 2019), or learnable augmentation search strategies (Cubuk et al., 2019; Lim et al., 2019; Hataya et al., 2020; Zheng et al., 2021). While data augmentation during training time brings much benefit, the challenge would lie in the training cost and the difficulty, given the continually increasing size of the training dataset.

On the other hand, training time augmentation cannot solve the issue once for all. In general, we consider that the performance on in-distribution data as the standard accuracy, the performance on the out-of-distribution data as the robustness or the generalization. Specifically, we consider some *corruption* will occur at test time that is *unknown* a priori. Consequently, such kind of corruption cannot be explicitly learnt during training stage e.g. by adopting certain data augmentation that attempts to explore the unknown data distribution.

Test time augmentation (TTA) is defined as transforming samples before inference at test time. Conventional TTA always requires averaging multiple predictions over different augmented test samples to obtain a final prediction. The major performance gain of conventional TTA methods heavily lies

in the ensembling mechanism (Lakshminarayanan et al., 2017), which inevitably requires multiple forward passes of the inference model. Recent studies on learnable TTA methods put more focus on how to select the best transformation policies at each inference, i.e. the one supposed to provide the largest performance gain compared to no transformation (Kim et al., 2020; Chun et al., 2022). By adopting instance-level transformation policies, these methods show significant improvement for corrupted (out-of-distribution) data without harm on clean (in-distribution) data. However, there exist still several limitations: (i) most methods still require model ensembling on different predictions to achieve the best performance; (ii) the desired iteration number of transformation before the inference is proportional to the cost of the transformation predictor, which limits the variety of the transformation; (iii) the transformation policy search is still under-explored thus leads to sub-optimal performance.

In this paper, we propose a cascade loss prediction method that, for the first time, only requires a single forward pass of the transformation predictor, while can output multiple desirable transformations iteratively. Our contribution can be summarized as follows:

- a novel cascade test time augmentation with sequential predictions by a single forward pass.
- a better trade-off between target model performance and inference cost with the first compatibility and analysis on various network architectures.
- a better exploration on the test data space which leads to state-of-the-art performance against various corruption benchmark.

## 2 RELATED WORKS

**General Data Augmentation**    Traditional data augmentation aims at enlarging training datasets to improve predictive performance. Recent works explore more diverse strategies of data augmentation such as by mixing up the features and their corresponding labels (Zhang et al., 2018), by cutting out some random certain area of mixed samples (DeVries & Taylor, 2017), or by cutting out then mixing up those samples with different strategies (Yun et al., 2019; Han et al., 2022). On the other hand, there are some studies on trainable augmentation policy (Cubuk et al., 2019; Lim et al., 2019; Hataya et al., 2020; Zheng et al., 2021). They focus rather on the exploration of larger data space and the automatic learning strategy for efficient training. These techniques are commonly used in many state-of-the-art models for their benefit on both accuracy and calibrations, bringing performance gain on standard benchmarks such as CIFAR (Krizhevsky et al., 2009) and ImageNet (Deng et al., 2009).

**Out-of-Distribution Robustness**    Sufficient augmentation is also a successful practice to improve out-of-distribution robustness. Hendrycks & Dietterich (2018) built the first benchmark for evaluating model robustness given different image corruption at test time. Hendrycks et al. (2019) proposed a simple data processing method to improve robustness; it augments training samples by mixing weighted random transformation operations and learns a distribution similarity between the original samples and the augmented samples. Wen et al. (2020) argued that simple model ensembles on top with such augmentation will degrade the performance, and then proposed a improved variant that dismisses the ones with high uncertainty. Zhang et al. (2021) proposed to adapt the model parameters by minimizing the entropy of the model's average output distribution across the augmentations, at test time. Whereas the inference becomes expensive due to its augmentation and adaptation procedure, thus limits the usability for other models or tasks.

**Test Time Augmentation**    Given a trained model, conventional test time augmentation is often carried out together with model ensembling, that is at inference with different augmented test samples, such as the conventional transformations e.g. cropping or flipping. Lyzhov et al. (2020) demonstrated that test time augmentation policies can be learned and introduced a greedy method for learning a policy of test time augmentation. Shanmugam et al. (2021) analyzed when and why test time augmentation works and presented a learning-based method for aggregating test time augmentations. Kim et al. (2020) selected suitable transformations for a test input based on their proposed loss predictor; without high additional computational cost, it carried out instance-level transformation at inference for the first time. However, the proposed method only explores one single trans-

formation for each sample, while requires model ensembles to achieve the performance on target model with multiple transformations. Recently, Chun et al. (2022) proposed a cyclic search for suitable transformations with the use of the entropy weight method, thus extend the instance-level augmentation to a larger data space. Whereas, the proposed cyclic mechanism is carried out on the entire loss predictor thus the model size is limited to be lightweight so as to achieve reasonable applicability.

## 3 METHOD

Loss prediction to find suitable transformations is an efficient search policy for test time augmentation (Kim et al., 2020; Chun et al., 2022). In this section, we introduce our method of a cascade loss prediction that outputs a succession of multiple transformations at a stretch. To begin with, we describe the general test time augmentation and the loss prediction pipeline in Section 3.1. Then, our Cascade-TTA method is in detail explained in Section 3.2. The cascade style contributes to the flexibility as well as the advantage in terms of calculation cost. Particularly in Section 3.3, we introduce the training method for our cascade loss predictor.

### 3.1 LOSS PREDICTION

Given a trained target model $\Theta_{target}$ and an input image $x$, the predictive result is based on the output $y$ of the target model:

$$y = \Theta_{target}(x). \tag{1}$$

Let $\mathscr{T} = \{\mathcal{T}_1, \mathcal{T}_2, ..., \mathcal{T}_k\}$ denote the candidate set of augmentations, test time augmentation is then conducted as:

$$y_{tta} = \frac{1}{k} \sum_{i=1}^{k} \Theta_{target}(\mathcal{T}_i(x)), \tag{2}$$

where $k$ indicates the size for ensemble effect. Conventional TTA methods usually carry out straightforward transformation for $\mathcal{T}_i$ such as cropping and flipping (He et al., 2016; Krizhevsky et al., 2017), indiscriminatingly performed on every input image. Assume we have an $N$-sized pre-defined transformation operation set, $T = \{t_1, t_2, ..., t_N\}$. Especially, $t_{\text{id}}$ is included in $T$ as `identity` to indicate no transformation operation is done. In conventional TTA, only one transformation is adopted each time on $x$, thus the size of the candidate set is $|\mathscr{T}| = N$.

In this paper, we define that each $\mathcal{T}_i$ is a sequence of independent augmentations as

$$\mathcal{T}_i = [\tau_{i_1}, \tau_{i_2}, ..., \tau_{i_L}], \tau_{i_j} \in T, \tag{3}$$

where $L$ is the iteration number of transformations, $L \geq 1$; $\tau$ is *one* single transformation from the pre-defined space. Thus conventional TTA with $L = 1$ is a special case by this definition. In general, the transformation space is $|\mathscr{T}| = N^L - N^{L-1} + 1$.

Following Kim et al. (2020), we propose to find the optimal $\mathcal{T}_i$ in an instance-aware manner:

$$\tilde{\mathcal{T}} = [\tilde{\tau}_1, \tilde{\tau}_2, ..., \tilde{\tau}_L] = f_{tta}(x), \tilde{\tau}_j \in T, \tag{4}$$

where $f_{tta}$ stands for the learned search criteria.

Given a trained $\Theta_{target}$, when $L = 1$, the loss value on augmented samples $\mathcal{L}_t(\Theta_{target}(\tau(x)), \hat{y})$ can signify the quality of the transformations $\tau$ (Kim et al., 2020), where $\hat{y}$ indicates the ground-truth of $x$ for the target model. Thus, the selection on $\tau$ is straightforward with the exact loss values. The loss predictor, denoted by $\Theta_{lp}$, can be trained independently in order to estimate the loss values corresponding to each pre-defined transformation (Kim et al., 2020):

$$f_{tta} \triangleq \Theta_{lp} \mid_{L=1}. \tag{5}$$

The loss predictor takes charge of telling by which transformations the target model achieves best performance. Since the output of loss predictor represents the quality ranking of the transformations, the benefit from ensemble effect is also possible. A cyclic version of the loss predictor has been proposed to deal with severely corrupted test samples (Chun et al., 2022), namely in the case of

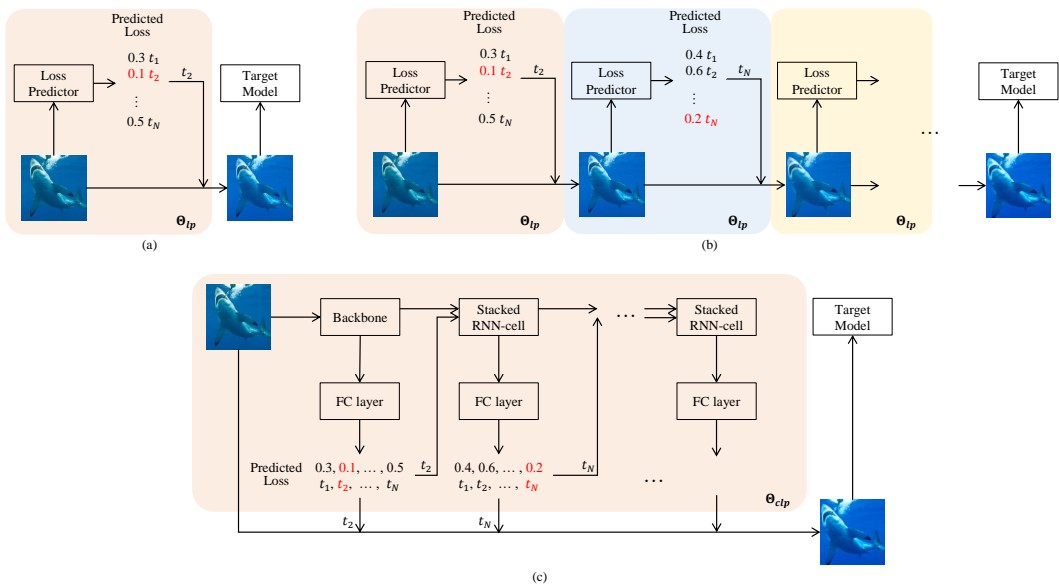

Figure 1: Illustration of different loss predictors at inference stage. The selection from pre-defined transformations is based on the assumption that lower predicted loss value corresponds to better transformation. (a) The single loss predictor for the best transformation when $L = 1$, selecting from $T = \{t_1, t_2, ..., t_N\}$. (b) The cyclic version of the loss predictor. $L$ steps of the single loss predictor form a cycle and produce $L$-sequenced transformations. (c) The cascade loss predictor, requiring only a single of forward pass prediction with $L$ transformations outputted. The backbone is used once while the stacked RNN-cell and the FC layer produce the outputs iteratively.

$L > 1$. Multiple repeated usage of the loss predictor forms a cycle, deeming the transformed image again as an input:

$$f_{tta} \triangleq \overbrace{\Theta_{lp}(\Theta_{lp}(\cdots\Theta_{lp}\cdots))}^{L \text{ times}} = \Theta_{lp}{}^L. \tag{6}$$

In the cyclic process, the loss predictor takes the intermediate transformed images in each iteration.

## 3.2 CASCADE LOSS PREDICTION

Multiple iterative transformations on a single test sample improves the potential of TTA. With $L > 1$, more operations are adequately performed to be better appropriate for the target model. Different from simple and plain repetition on the loss predictor, our method focusses on how to produce a succession of transformations once with a single network:

$$f_{tta} \triangleq \Theta_{clp}, \tag{7}$$

where $\Theta_{clp}$ is our novel cascade loss predictor performing merely once with no limitation on $L$.

As the ensemble is simple to implement in practical, we will take $k = 1$ in the sequel for simplicity. Figure 1 shows the overview of the single loss predictor (Kim et al., 2020), the cyclic version (Chun et al., 2022) and our cascade loss predictor respectively. As noted earlier, the single implementation only caters for $L = 1$ and the cyclic version just calls the loss predictor multiple times block-wise.

In this paper, we propose a cascade architecture as shown in Figure 1(c). It uses recurrent neural network (RNN) to capture the semantic information of the transformed image in each iteration, and realizes predicting iterative transformations with no need to take advantage of the intermediate transformed images. Significantly, only a single forward pass of the cascade loss predictor is required to obtain $L$ desired transformations iteratively. Without the tedious process of re-inputting the transformed image into the loss predictor, the proposed cascade network just accepts once the original input $x$ but provides a succession of appropriate transformations. On this occasion, we are

---

**Algorithm 1** Inference of our cascade predictor

---

**Inputs:** An input test image $x_1$
**Output:** A succession of transformations $\mathcal{T}$
 1: $h_1 \leftarrow \mathbf{Backbone}(x_1)$
 2: $\tau_1 \leftarrow \arg\min(\mathbf{FC}(h_1))$
 3: $\mathcal{T} \leftarrow [\tau_1]$
 4: **for** each $i \in [2, L]$ **do**
 5:     **if** $\tau_{i-1}$ is $t_{\mathrm{id}}$ **then**
 6:         break
 7:     **end if**
 8:     $h_i \leftarrow \mathbf{RNN}(h_{i-1}, g(\tau_{i-1}))$
 9:     $\tau_i \leftarrow \arg\min(\mathbf{FC}(h_i))$
10:     $\mathcal{T} \leftarrow \mathcal{T} + [\tau_i]$
11: **end for**

---

able to directly perform the obtained $L$-sequenced transformation $\mathcal{T} = [\tau_1, \tau_2, ..., \tau_L]$ at a stretch, and straightforward get the final augmented sample $\mathcal{T}(x)$ that should be fed into the target model. None of the intermediate transformed images are substantially utilized for the cascade network.

The proposed cascade loss predictor consists of three parts including the backbone, the stacked RNN-cell and the FC layer. The direct and concise prediction on the successive transformations comes from the RNN architecture of the cascade network. In other words, the stacked RNN-cells process the dependencies through the cascade loss predictions. Inspired by the tremendous success of RNN models in sequence processing (Bahdanau et al., 2014; Yang et al., 2018), we put forward a reasonable RNN-based loss predictor for sequential transformations generation.

We present the inference procedure of the cascade predictor as shown in Algorithm 1. $h_i$ denotes the feature of state at each iteration. In the first iteration, we take advantage of the backbone feature $h_1$ for transformation prediction. From the second iteration, instead of using the explicitly augmented sample i.e. $x_i = \tau_{i-1}(\tau_{i-2}(\ldots \tau_1(x_1)\ldots))$, we apply the iterative hidden state as following:

$$h_i = \mathbf{RNN}(h_{i-1}, g(\tau_{i-1})), \tag{8}$$

where $g$ is the embedding network to embed the transformations to a feature space. Then the optimal $\tau_i$ at this iteration is selected by the minimum predicted loss value from a linear regressor. We propose to use two stopping criteria: (i) $t_{\mathrm{id}}$ is achieved; (ii) maximum iteration $L$. See Appendix A for more details on the procedure of the cascade prediction.

For efficiency, the EfficientNet-B0 (Tan & Le, 2019) with modification (Yoo & Kweon, 2019) is often used as the backbone of the loss predictor (Kim et al., 2020; Chun et al., 2022), whose cost is relatively negligible to the target model. The downsizing operation into 64 by 64 pixels for ImageNet dataset (Deng et al., 2009) further reduces the computation. However, light backbones intuitively lack of representative ability, especially for the multiple iterative predictions. As the complexity of backbone increases, it is also important to consider the trade-off between the loss prediction cost and the improved performance. We will show in Section 4.3 the analysis and experimental results on different complexity of backbones for the cascade loss predictor.

### 3.3 PRACTICAL TRAINING STRATEGY

We illustrate the training method of Cascade-TTA in Figure 2. Given a training image $x_1$, the ground-truth is defined as $L$ vectors corresponding to $L$ training iterations. Specifically, each of the vectors consists of $N$ loss values, which is calculated from the target model regarding augmented images as inputs by the pre-defined transformations. For iteration $i$, we note the ground-truth loss value vector as:

$$\mathbf{l}_i = \{\mathcal{L}_t(\Theta_{target}(t_j(x_i)), \hat{y})\}, j \in [1, N], \tag{9}$$

where $x_i$ is the intermediate image output from the last iteration and $t_j(x_i)$ indicates the transformed one. Note that $\mathcal{L}_t$ indicates the loss function for the target model which has been all fixed. Different from the inference stage, $x_i$ needs to be explicitly produced for the ground-truth generation.

In practice, the intermediate image $x_i$ for iteration $i$ is augmented by the pre-defined transformations $T = \{t_1, t_2, ..., t_N\}$, and each of the transformed image $t_j(x_i)$ is in parallel fed into the target model

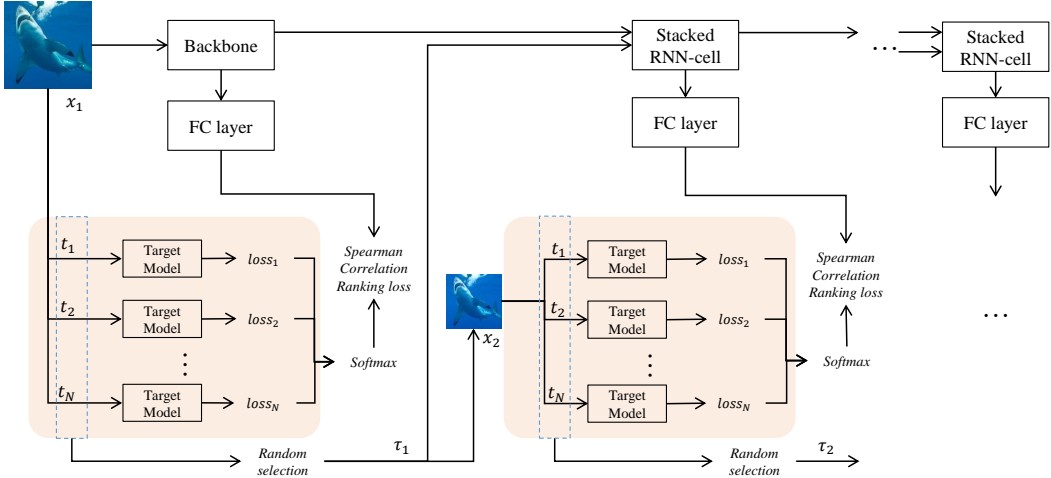

Figure 2: Illustration of the training procedure of the cascade loss predictor. The shaded area is the single ground-truth generator, which is iterated several times. Note that even when training the predictor, the target model still remains fixed.

to obtain the loss value $\mathcal{L}_t(\Theta_{target}(t_j(x_i)), \hat{y})$. Before training the loss predictor, loss values are generated from the trained target model by using $\mathcal{L}_t$, the loss function of the target model at its training stage. The generated loss values are then gathered to produce $\mathbf{l}_i$ and processed by softmax function. Concretely, in the first iteration, the training sample $x_1$ is transformed to $\{t_j(x_1)\}$, which are fed into the target model for loss value calculation and eventually contribute to the first ground-truth vector $\mathbf{l}_1$. For the diversity and balance of training data, we randomly assign a transformation $\tau_i$ (except $t_{\mathrm{id}}$) for the current image $x_i$. For instance, a random transformation $\tau_1 \in T$ is assigned for $x_1$ at the end of the first iteration, so the intermediate image for the second iteration is defined as $x_2 = \tau_1(x_1)$. During the second iteration, similarly the target model takes the transformed $\{t_j(x_2)\}$ and produces the second ground-truth vector $\mathbf{l}_2$. And so on, for each of the iterations, sequentially we can get all of the ground-truth vectors $\{\mathbf{l}_i\}$. On the other hand, a vector of predicted loss values is produced by the FC layer in each iteration, so our cascade loss predictor optimizes the correlation between the predicted and ground-truth vectors in all training iterations.

The training iteration number can be determined independent of the test time iteration number. Spearman correlation ranking loss (Engilberge et al., 2019) is used for optimization in all iterations together, which is a better description of the transformation quality than the exact ones. We regard the iteration number as a part of batch size during training the cascade loss predictor.

## 4 EXPERIMENTAL RESULTS

### 4.1 CIFAR100

CIFAR-100 benchmark (Krizhevsky et al., 2009) is a widely-used classification dataset. A total of 60000 images with 32 by 32 pixels belong to 100 classes. The corruption version of CIFAR-100-C (Hendrycks & Dietterich, 2018) is introduced for evaluation, and the corrupted variant consists of a total of 19 kinds with 5 severities. The *error* rate for clean data and the average corruption *error*, $CE_c = \frac{1}{5}\sum_{s=1}^{5} E_{c,s}$ define the evaluation criteria, where $E_{c,s}$ denotes the Top-1 *error* rate on corruption $c$ with severity $s$. The pre-defined transformations used in all of our experiments consist of 12 kinds of operations. See Appendix B for details.

As shown in Table 1, experiments are conducted on the comparison between Cascade-TTA and existing TTA methods. Results for both clean and corrupted data are presented. The two target models that we use are both augmented with AugMix (Hendrycks et al., 2019) with different architecture of Wide-ResNet-40-2 (Zagoruyko & Komodakis, 2016) and ResNeXt-29 (Xie et al., 2017). Due

Table 1: Evaluation result on CIFAR-100(-C) dataset. Metric for corrupted set is average corruption *error*. The Cascade-TTA results are shown with $L = 3$.

| Target Model | TTA method | Target Model Cost | Clean↓ | Corrupt↓ |
|---|---|---|---|---|
| Wide-ResNet | Center-Crop | 1 | 23.00 | 35.34 |
| | Horizontal-Flip | 2 | 22.36 | 34.38 |
| | 5-Crops | 5 | 22.97 | 35.16 |
| | Random-TTA | 1 | 27.86 | 40.89 |
| | Cascade-TTA | 1 | 23.08 | 34.12 |
| ResNext | Center-Crop | 1 | 20.41 | 33.51 |
| | Horizontal-Flip | 2 | 19.82 | 32.90 |
| | 5-Crops | 5 | 20.11 | 33.26 |
| | Random-TTA | 1 | 25.51 | 38.94 |
| | Cascade-TTA | 1 | 20.44 | 31.99 |

to the low resolution of CIFAR-100 images, we simply use modified EffcientNet-B0 as the backbone of the cascade loss predictor. The conventional augmentation strategies such as Center-Crop, Horizontal-Flip and 5-Crops have always been used in previous applications. In spite of the multiple target model cost for ensemble in Horizontal-Flip and 5-Crops, the improved performance is still limited. Random-TTA means to choose a random transformation out of the pre-defined augmentations. The degraded performance indicates the sufficiency of the transformation diversity. Our proposed Cascade-TTA outperforms not only the methods with same computation cost, but also outperforms the methods with larger cost of target model.

## 4.2 IMAGENET

ILSVRC 2012 classification benchmark (ImageNet) (Deng et al., 2009) consists of 1.2 million images of 1000 classes. The corrupted variant of ImageNet-C (Hendrycks & Dietterich, 2018) is also used for out-of-distribution evaluation. The *error* rate for clean data and the mean corruption *error* ($mCE$) metric for corrupted data is calculated for evaluation criteria. Table 2 shows the performance of different TTA methods on ResNet-50, but with different train-time augmentation. Here we also implement two backbones of the loss predictor with different complexity, EfficientNet-B0 and ResNet-50. For target models trained with Standard, our performance by a clear margin exceeds the single version of the loss predictor, no matter which the backbone is used. Additionally, ensemble of two transformed images by Cascade-TTA with multiple cost of target model, produces lower error rate up to expectations. For target models trained with AugMix, Cascade-TTA also achieves best performance on both of the backbones.

## 4.3 BACKBONE VS PERFORMANCE

We also explore on various network architectures to dig into the trade-off between backbone complexity and performance. Especially, we show in Figure 3 the calculation cost on different iteration when using an EfficientNet-B0 or a ResNet-50. As we can see, there exists essential difference on the calculation cost between Cyclic TTA (Chun et al., 2022) and our Cascade-TTA. With the repeated usage of the loss predictor for Cyclic TTA, the cost explicitly multiplies, greatly requiring $L$ times of the backbone cost. For our proposed Cascade-TTA, the cost on contrary mainly depends on the stacked RNN-cell, which concretely contains one time of the backbone and $L - 1$ times of the stacked RNN-cell. It is important to note that the stacked RNN-cell is generally light enough to rival EfficientNet-B0.

We carry out experiments with diverse backbones to explore the relation between the complexity of backbone and the improved performance. Table 3 shows the performance of Cascade-TTA with 6 kinds of backbones for cascade loss predictor, ranging from EfficientNet-B0 to ShuffleNetv2 (Ma et al., 2018) and ResNet-50. The input resolution is adaptively adjusted as 64 by 64 pixels on each EfficientNet family backbone. We choose to use the trained target model with ResNet-50 as backbone and Standard as training data augmentation. Meanwhile, results of different maximum iteration numbers are presented in proper order for compare. Experiments show that with light backbones, short length of iteration can efficiently improve the performance, but the increasing

Table 2: Evaluation result on ImageNet(-C) dataset with target model as ResNet-50. The Cascade-TTA results are shown with $L = 2$.

| Train-Aug | TTA method | Target Model Cost | Clean↓ | Corrupt↓ |
|-----------|------------|-------------------|--------|----------|
| | Center-Crop | 1 | 24.14 | 77.54 |
| | Horizontal-Flip | 2 | 23.76 | 76.50 |
| | 5-Crops | 5 | 23.57 | 76.08 |
| | Random-TTA | 1 | 26.82 | 81.55 |
| Standard | EfiicientNet-B0 Single-TTA* | 1 | 24.19 | 75.09 |
| | EfiicientNet-B0 Cascade-TTA | 1 | 24.17 | 74.20 |
| | EfiicientNet-B0 Cascade-TTA | 2 | 24.16 | 74.05 |
| | ResNet-50 Single-TTA* | 1 | 24.16 | 74.46 |
| | ResNet-50 Cascade-TTA | 1 | 24.20 | 74.33 |
| | ResNet-50 Cascade-TTA | 2 | 24.14 | 74.02 |
| | Center-Crop | 1 | 22.39 | 66.57 |
| | Horizontal-Flip | 2 | 22.14 | 65.84 |
| | 5-Crops | 5 | 21.70 | 65.02 |
| AugMix | Random-TTA | 1 | 24.15 | 70.58 |
| | EfiicientNet-B0 Cascade-TTA | 1 | 22.37 | 64.87 |
| | ResNet-50 Cascade-TTA | 1 | 22.38 | 64.49 |

[1] Single-TTA* indicates the work of Kim et al. (2020).
[2] We re-implemented Cyclic TTA but did not receive results as Chun et al. (2022) presents.

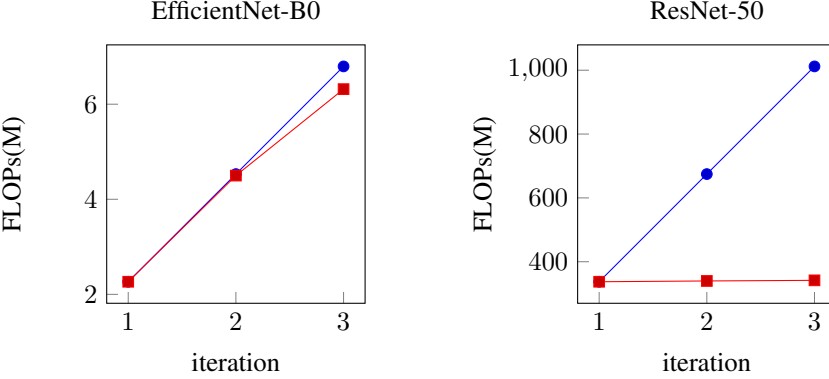

Figure 3: **Iteration Number vs. Calculation Cost.** The blue line stands for the usage of Cyclic TTA while the red line represents our method. Left: EfficientNet-B0 as backbone, the cost of Cyclic TTA is marginally more costy. Right: ResNet-50 as backbone, the cost increases sharply along with the iteration number in Cyclic TTA while our Cascade-TTA is almost impervious.

iterations soon start to depress the improvement even if the cost of predictor expands. Therefore, the light backbones for the loss predictor limit the improvement from the iteration number. We assume that it is due to the insufficient representation ability for the growing iterations of prediction. However, there is no such issue when using large backbones. The increase on iteration stability provides greater benefits, and the extra cost from RNN-cell is still light. Eventually the improvement exceeds light backbones with sufficient iteration. In addition, experimental results show that the largest backbone as ResNet-50 provides the best performance with long enough iterations. As a consequence, when desiring optimal performance gain out of the loss predictor, long iteration and large backbone offer the best alternative. In this case, Cascade-TTA compared with Cyclic TTA, requires merely the extra cost of the light RNN-cells instead of the multiple large backbones, so Cascade-TTA provides the best trade-off with significant improvement.

Table 3: The ImageNet-C results of Cascade-TTA on backbones with different complexity. The first, second, and third iteration of results are shown for the trend.

| TTA backbone | FLOPs(M) | | | Clean | | | Corrupt | | |
|---|---|---|---|---|---|---|---|---|---|
| | $L=1$ | $L=2$ | $L=3$ | $L=1$ | $L=2$ | $L=3$ | $L=1$ | $L=2$ | $L=3$ |
| EfficientNet-B0 | 2.265 | 4.498 | 6.317 | 24.17 | 24.17 | 24.18 | 74.70 | 74.20 | 74.73 |
| EfficientNet-B2 | 3.273 | 5.590 | 7.452 | 24.19 | 24.20 | 24.20 | 74.48 | 74.32 | 75.66 |
| EfficientNet-B4 | 5.674 | 8.247 | 10.240 | 24.16 | 24.18 | 24.18 | 74.66 | 74.49 | 75.07 |
| EfficientNet-B8 | 16.330 | 19.581 | 21.922 | 24.21 | 24.21 | 24.20 | 74.61 | 74.66 | 76.13 |
| ShuffleNetv2 | 48.692 | 51.152 | 53.087 | 24.19 | 24.18 | 24.18 | 74.80 | 74.46 | 74.19 |
| ResNet-50 | 337.308 | 339.767 | 341.702 | 24.20 | 24.20 | 24.20 | 74.64 | 74.33 | 74.05 |

## 5 DISCUSSION AND CONCLUSION

The visualization results are shown in Figure 4. The images are transformed iteration by iteration and eventually are correctly classified by the target model.

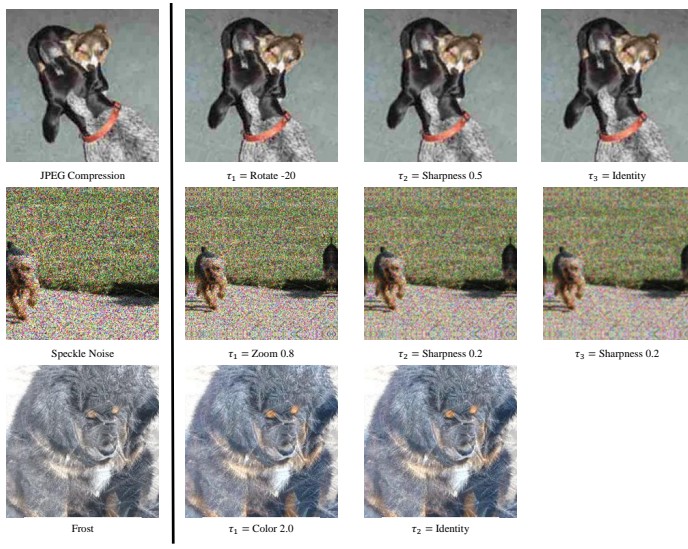

Figure 4: Visualization of selected samples on ImageNet when $L = 3$. First column: corrupted images from ImageNet-C; Second to fourth column: transformed images performed by our method iteratively. The corruption way or the selected transformation is annotated below each image.

To conclude, in this paper, we propose a novel test time augmentation using a cascade loss prediction. For the first time, multiple transformations can be predicted iteratively with one single forward pass of the predictor. The cascade predictor is computational efficient and compatible to various network architectures with limited additional cost, thus holds a promising applicability. Due to the fact that the training space is exponential to the pre-defined type of transformations, we propose a practical training strategy to train the proposed cascade predictors. Experimental results validate the effectiveness of the proposed method. We suppose that enlarging the pre-defined transformation space could further upgrade the performance, while an efficient training strategy is essential, which could be expected as a future work.

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

## A    DETAILED ON THE CASCADE PREDICTION

Here we explain more details on how the cascade prediction works.

As shown in Algorithm 1, let $x_1$ be a test sample. In the very first iteration, the cascade loss predictor takes $x_1$ without transformations as input and produces $h_1$ out of the backbone. Here we define the last layer of the backbone as a pooling layer, so $h_1$ is a vector representing the feature of $x_1$. Then the FC layer takes $h_1$ to perform the loss prediction, and selects the most appropriate transformation as $\tau_1$. On the other hand, the stacked RNN-cell comes into play from the second iteration. Instead of fetching the transformed $x_2 = \tau_1(x_1)$ substantially, the stacked RNN-cell takes $h_1$ as the hidden state and the embedded $\tau_1$ as input. Here we embed the transformation by learning a vector to represent it. After the two representations are processed by the stacked RNN-cell, the feature of $x_2$ can be still obtained as the output $h_2$, without any processing by the backbone repeatedly. The FC layer is called again to perform the second time of loss prediction, taking $h_2$ as input and selecting $\tau_2$ as output. As we particularly employ the stacked RNN-cell instead of a single cell, the vector $h_1$ is cut into several segments and respectively put into the single cells at different levels. The FC layer re-concatenates the several hidden states before the linear. In doing so, the deducible cascade fashion comes into existence. Aside from the old-fashioned loss prediction in the first iteration, for each iteration $i$, the hidden state $h_{i-1}$ and the embedded selected transformation $\tau_{i-1}$ from the last iteration are inputted into the stacked RNN-cell, to provide the current hidden state $h_i$. And then the FC layer is responsible to predict the appropriate transformation $\tau_i$ from $h_i$. Each state of $h_i$ simulates the feature of $x_i = \tau_{i-1}(\tau_{i-2}(\ldots \tau_1(x_1)\ldots))$. Eventually the cascade loss predictor is implemented continuously until two exit signals. One is the predicted best transformation as `identity` and the other is the predetermined ceiling iteration number $L$. The former indicates the best condition of current image while the latter prevents endless prediction.

RNN as a kind of network with feedback loops, dynamically connects last output with current input, which greatly matches the searching process for iterative transformations. Given the ground-truth label of $x_1$ with $L$ iterations, $\mathcal{T} = [\tau_1, \tau_2, \ldots\ldots, \tau_L]$, during $i^{th}$ iteration, the cascade loss

predictor aims at finding $\tau_i$. The determining factor of $\tau_i$ lies in the loss values of transformed $x_i = \tau_{i-1}(x_{i-1})$, while $\tau_{i-1}$ is corresponding to the prediction in the last iteration thus $\tau_{i-1}$ is supposed to be inputted into the current iteration. In addition, RNN can memorize previous states of $x_{i-1}$. Therefore, the utilization of RNN in cascade loss prediction reasonably finds appropriate transformations iteratively.

## B    PRE-DEFINED TRANSFORMATIONS

We use pre-defined transformations to a great extend following Kim et al. (2020); Chun et al. (2022). With a bit difference on the magnitudes for better performance on iterative prediction, there are a total of 12 kinds of pre-defined transformations in our experiments, including: identity, rotation, zoom, contrast, sharpness and saturation. Rotation refers to rotating the image 20 or -20 degrees to the center point; Zoom refers to resizing the image to 0.8 or 1.2 times and then cropping for the original size; Contrast refers to adaptively streching the image to uniform contrast level; Sharpness refers to adjusting the sharpness of the image to 0.2, 0.5, 3.0 or 4.0; Saturation refers to changing the color saturation of the image to 0.5 or 2.0.

