# OpenReview forum: "Learning Test Time Augmentation with Cascade Loss Prediction"
_ICLR.cc/2023/Conference — Submitted to ICLR 2023_

### Official Review · Reviewer_N36J · 2022-10-13

**Confidence:** 3
**Correctness:** 3
**Technical Novelty And Significance:** 2
**Empirical Novelty And Significance:** 2
**Recommendation:** 3

**Clarity, Quality, Novelty And Reproducibility:**

I believe that the presentation of this paper is clear, and enough details have been provided to reproduce this method. However, I have concerns about its novelty.

**Strength And Weaknesses:**

Strength:
1. This paper is well-written and easy to follow
2. This paper is well-motivated and the motivation of this paper is reasonable.

Weaknesses:
1. My first concern w.r.t. this paper lies in its limited novelty. While I understand that compared with the previous method that uses either the simple version or the cyclic version of the loss predictor, this paper does demonstrate its advantages. However, it seems to me that this paper technically just applies RNN which is not a new technique already. Thus, while it is reasonable to choose to use RNN here, from my perspective, this does not bring enough novelty as no specific designs have been made for the task of TTA.
2. I am also a bit confused about why the authors suddenly jumped to the usage of ResNet-50. Specifically, intuitively, I think that the task of the loss predictor is not a very hard task. Thus, the usage of a lightweight backbone EfficientNet-B0 is adequate and reasonable from my perspective. Thus, while the authors do demonstrate that their method can be much more efficient than cyclic predictor with ResNet-50 as the backbone, I am a bit confused why ResNet-50 should be used as the backbone in the first place. Note that the improvement of efficiency is quite small on EfficientNet-B0, and it seems from the experimental section, the usage of ResNet-50 does not lead to a significant performance enhancement compared to EfficientNet-B0.
3. At last, I suggest the authors make their evaluation metric more clearly in the experiment section. Specifically, the name of the metric and whether a small or a large number leads to better performance can be included in the table. The current title of the table with only "Clean" and "Corrupt" there brings me a slight reading difficulty.

**Summary Of The Paper:**

To cope with the task of TTA, this paper proposes to replace the simple version or the cyclic version of the loss predictor with a cascaded version utilizing an RNN architecture. The authors evaluate their method on CIFAR-100 and ImageNet.

**Summary Of The Review:**

Overall, I have concerns about the novelty of this paper. Besides, while I agree that this paper is more efficient than the cyclic version, it seems that this efficiency is only significant when a large model is used as the backbone. However, it seems that a small model as the backbone is already enough in this task.

---

> ### Author Response · Authors · 2022-11-18
> **Response to Reviewer N36J**
>
> Please refer to ***General Response*** for the questions about **novelty(Q1)**.
>
> **Q9: ResNet-50**
>
> **A9**: The reason why we turn to choose ResNet-50 as backbone instead of lighter Efficient-B0 is in Section 4.3. According to the results in Table 3, when we use EfficientNet-b0 as backbone, the results are worse as the testing iteration increases. We consider it resulting from the weak representation ability of the light backbone. The features cannot be presented exactly when the RNN-cells are cascaded by large iteration. Therefore, we assume that a heavier backbone whose representation ability is stronger should help. The results shown in Table 3 prove our assumption. When ResNet-50 is used, the results from iteration 3 do not get worse any longer (like EfficientNet-B0, EfficientNet-B2 and EfficientNet-B4), but keep getting better. In the revised paper, we move the last paragraph in Section 3.2 to Section 4.3 to make the structure more logical and well-reasoned.
>
> **Q10: Evaluation metric**
>
> **A10**: We apologize for the unclear explanation. We add more description about the evaluation metric in Section 4. Please refer to the revised paper.

---

### Official Review · Reviewer_yeBc · 2022-10-24

**Confidence:** 4
**Correctness:** 3
**Technical Novelty And Significance:** 2
**Empirical Novelty And Significance:** 2
**Recommendation:** 3

**Clarity, Quality, Novelty And Reproducibility:**

* Clarity:4
* Quality:3
* Novelty:3
* Reproducibility:3


**Strength And Weaknesses:**

Strength
The idea of using RNN seems natural and useful..
The computational cost of the proposed method is showed impervious with the iteration number on Resnet-50.
The paper is well-organized and easy to follow

Weakness
The experiments are not convincing enough. The paper does not compare with any SOTA methods listed in the related works.
As the results show in Table2, the proposed method has no obvious advantage over Single-TTA.
It's necessary to compare the evaluation results and the calculation cost simultaneously to get a convincing conclusion, while both of them are insufficient in this paper.


**Summary Of The Paper:**

The paper proposes a test time augmentation using a cascade loss prediction method which only requires a single forward pass of the transformation predictor to select multiple transformations. In contrast to the repeated usage of one loss predictor in cyclic-TTA, the proposed method uses RNN to capture the semantic information in each iteration and predict transformations without intermediate images. Experiments are conducted on various architectures to explain the trade-off between computational cost and model performance.

**Summary Of The Review:**

Overall, the main idea of this paper is interesting but the experiments are not convincing. Therefore, I suggest a rejection for this paper unless more experiments are complemented.

---

> ### Author Response · Authors · 2022-11-18
> **Response to Reviewer yeBc**
>
> Please refer to ***General Response*** for the questions about **novelty(Q1)**, **lack of comparison(Q2)** and **insufficient improvement(Q3)**.
>
> Hope our response could address your concern.

---

### Official Review · Reviewer_2Tn9 · 2022-10-25

**Confidence:** 4
**Correctness:** 3
**Technical Novelty And Significance:** 1
**Empirical Novelty And Significance:** 2
**Recommendation:** 3

**Clarity, Quality, Novelty And Reproducibility:**

Quality
---

I expected to see a comparison to cyclic TTA in the experiments, but as far as I can tell, that is missing. The experiments do seem to show some improvements compared to other baselines, but the improvements are generally small and it is unclear how competitive these baselines truly are.

Clarity
---

I do not understand why the authors refer to their method with the term "cascade" when simply referring to it as RNN-based would be much simpler and clearer. Eq (2) seems to only make sense if y is a real value, which it is not in classification. I was not able to follow Section 3.3 detailing the training strategy, and given that there is still room in the main paper, I would suggest focusing on providing more detail in that section to help clarify these important points. Generally, the vocabulary used throughout ("ascendancy", "rough", etc.) unnecessarily complicates the exposition and I would recommend being more consistent and rigorous with the writing.

Originality
---

It seems like the main novelty of the proposed method is in using an RNN to output sequential augmentations. This does not seem significantly novel by itself. As noted, cyclic TTA also outputs sequential augmentations but is more computationally expensive due to working in image space. This seems to lead to a natural comparison: cyclic TTA, but with the same backbone that embeds the image into the lower dimensional feature space. In general, comparisons such as these (and standard cyclic TTA) would go a long way in strengthening the empirical aspects of the paper, which are needed due to the lack of a significantly novel technical contribution.

**Strength And Weaknesses:**

Strengths
---

- Experiment seem to show that the method holds promise for achieving favorable performance and efficiency.
- The method is relatively simple and easy to understand.

Weaknesses
---

- The paper is difficult to follow in numerous places (see below).
- It is unclear whether the proposed idea is significant enough to warrant publication at ICLR.

**Summary Of The Paper:**

This paper proposes a method for learning data augmentations for improving test performance in the presence of distribution shift. The method uses an RNN based augmentation module rather than a feedforward module that may be run multiple times. Experiments show that this method seems promising on standard corrupted image benchmarks.

**Summary Of The Review:**

In summary, there are several important improvements that can be made as laid out above. In the paper's current state, I am recommending reject.

Edit after author response
---

Thank you for the response, and apologies for the late reply. My recommendation remains unchanged. It seems that the other reviewers and I agree that this paper will benefit significantly from another full reviewing round before publication.

---

> ### Author Response · Authors · 2022-11-18
> **Response to Reviewer 2Tn9**
>
> Please refer to ***General Response*** for the questions about **novelty(Q1)**, **lack of comparison(Q2)**, **insufficient improvement(Q3)** and **training details(Q4)**.
>
> **Q7: The term "cascade"**
>
> **A7**: Though our method is based on RNN, each RNN-cell in our architecture is regarded as a cascadable sub-network rather than a simple recurrent unit. For traditional usage of RNN, each cell is almost undifferentiated, representing information at some time point, just like a tile of chain. Different from it, the utility of RNN in our Cascade-TTA is progressive. The feature of the intermediate transformed images are extracted level by level. The term “cascade” can be more accurate to state the meaning. Specifically, when $L=1$, only the backbone and the FC layer are passed, which is a total different stage compared to $L\geq2$ with RNN related; so we consider that it is more reasonable to call our method as “cascade” rather than “RNN-based”.
>
> **Q8: Eq (2) and vocabulary**
>
> **A8**: We apologize for the confusing notation and we revise that $y$ is noted as the output logit by target model instead of the result. We also find other alternative words for the vocabularies you mentioned ("ascendancy", "rough", etc.). Please refer to the revised paper.

---

### Official Review · Reviewer_6hqy · 2022-10-25

**Confidence:** 3
**Correctness:** 3
**Technical Novelty And Significance:** 2
**Empirical Novelty And Significance:** 1
**Recommendation:** 3

**Clarity, Quality, Novelty And Reproducibility:**

The overall quality of this paper is not strong enough. There are some issues unclear to me. I don't know how to train the loss predictor and cannot reproduce the results after reading this paper.

**Strength And Weaknesses:**

Strength:
+ This paper presents a method, which is able to produce multiple data augmentations with single forward pass.

Weakness:
- The novelty of this work is not surprisingly strong. The proposed model is different to previous architecture, however, applying RNN to produce multiple output is a common method.

- From the experimental results, it is hard to observe clearly better results than other TTA methods, for example Horizontal-Flip.

- How do you train the loss predictor. This is unclear how to train a loss predictor on specific datasets.

- How to select the predefined candidate transformations. For individual instances, different transformations are required. How do you prepare the transformation set for the best performance for various datasets.

- Need citation after the first sentence in section 3.


**Summary Of The Paper:**

This paper presents a model for selecting test-time data augmentation to boost a classification model. The proposed method is based on RNN to gradually output multiple augmentations, selected from a predefined augmentation set. The idea is based on loss prediction for finding suitable augmentations. In each step of RNN prediction, the model outputs a predicted loss value and then select the best augmentation. To show the effectiveness of the proposed method, this paper conducts experiments on CIFAR-100 and ImageNet datasets, and compare this method with many other test-time augmentation strategies.

**Summary Of The Review:**

This paper does not have a clear merit to be accepted by ICLR. They present an architecture for test-time augmentation, however, no clear benefit from the proposed method can be achieved.

---

> ### Author Response · Authors · 2022-11-18
> **Response to Reviewer 6hqy**
>
> Please refer to ***General Response*** for the questions about **novelty(Q1)** and **training details(Q4)**.
>
> **Q5: Pre-defined augmentations**
>
> **A5**: We apologize for the lack of clearness on which augmentations are used in our experiments. Greatly following *Kim et al.*[1], our pre-defined augmentations consist of 12 operations (Identity, Rotate-20, Rotate20, Zoom0.8, Zoom 1.2, AutoContrast, Sharpness0.2, Sharpness0.5, Sharpness3.0, Sharpness4.0, Color0.5, Color2.0). For more detailed information, please refer to the added Appendix B in the revised paper.
>
> **Q6: Need citation after the first sentence in Section 3**
>
> **A6**: Thank you for the suggestion. We make modification in the revised paper.
>
> [1] : Kim, Ildoo, Younghoon Kim, and Sungwoong Kim. "Learning loss for test-time augmentation." Advances in Neural Information Processing Systems 33 (2020): 4163-4174.

---

### Author Response · Authors · 2022-11-18
**General Response**

We would like to greatly appreciate the comments from all the reviewers and have made corresponding modifications in the revised paper. Specifically, changes were made mainly in the following aspects: 1) More detailed information is described in Section 3.3 to help readers understand the training procedure of our method and reproduce the results; 2) Appendix B is added to explain the pre-defined transformations in our experiment; 3) The captions and some method names in tables are modified for better comprehension of the comparison on the experiment results; 4) The last paragraph in Section 3.2 is moved to Section 4.3 to make the structure more logical and well-reasoned. Additionally, we address some of your common concerns below, and reply to each reviewer point-to-point later.

 **Q1: Lack of novelty**

**A1**: The idea of using RNN is natural and useful (as Reviewer yeBc mentioned), while we argue that this simplicity is precisely one of the advantage of our method. Although RNN is an existing technology, how to effectively apply it for sequenced transformation prediction is not straightforward. Usually RNN memorizes time sequence information in previous works; however, we utilize it to simulate the semantic feature of transformed image. Further, our method is the first to model the process of the sequenced transformation by RNN features, including deeming image feature as hidden state and embedding transformations. Therefore, our work is a novel TTA method with sequential predictions under the help of RNN.


**Q2: Lack of comparison in experiments**

**A2**: We do re-implement the experiment results of *Kim et al.*[1], and we call their method as Single-TTA in Table 2. We apologize for the confusing name of the method and we revise our paper for clearer reference. Also, we have tried to reproduce the results of *Cyclic TTA*[2] by ourselves due to the absence of open source code by them. Unfortunately, the results from the paper[2] is not reproducible, which is currently mentioned in the footnote of Table 2 for more clearness.

**Q3: Insufficient improvement**

**A3**: Following *Kim et al.*[1] and *Cyclic TTA*[2], we evaluate our method using error rate on clean data and on corrupted data. For one thing, the error rate of clean data nearly keeps the same as Center-Crop, indicating our Cascade-TTA has no harm for clean image. More importantly, the error rate of corrupted data does drop, no matter comparing with conventional TTA methods or *Kim et al.*[1]. Additionally, Horizontal-Flip or 5-Crops need more cost for the target model while ours can do the trade-off. For instance, in Table 2, with Standard as Train-Aug and EfficientNet-B0 as backbone, taking 2 times for the target model cost, our method decreases the error rate by 2.45 points from Horizontal-Flip; Taking 1 time for the target model cost, our method decreases the error rate by 0.89 points from *Kim et al.*[1]. It is worth noting that both *Kim et al.*[1] and we utilize the loss prediction pipeline but our cascaded architecture can indeed improve the performance.

**Q4: Training details**

**A4**: We revise Section 3.3 for better understanding with more implementation details. Please refer to the revised paper.

[1]: Kim, Ildoo, Younghoon Kim, and Sungwoong Kim. "Learning loss for test-time augmentation." Advances in Neural Information Processing Systems 33 (2020): 4163-4174.

[2]: Chun, Sewhan, Jae Young Lee, and Junmo Kim. "Cyclic test time augmentation with entropy weight method." Uncertainty in Artificial Intelligence. PMLR, 2022.

---

### Decision · Program_Chairs · 2023-01-20

**Decision:**

Reject

**Justification For Why Not Higher Score:**

The paper is far below the acceptance threshold.


**Justification For Why Not Lower Score:**

N/A


**Metareview: Summary, Strengths And Weaknesses:**

We appreciate the authors for addressing our comments and concerns and revising certain parts of the paper to address some of them. While the simple method proposed is more efficient than related TTA methods proposed by others before, our general concern is that its novelty is not significant enough for publication in ICLR. Moreover, from the experiment results, its improvement over other TTA methods is also not very significant. We hope the authors find our comments useful to improve their work for future submission.